# Chimeric Antigen Receptor T Cell Bearing Herpes Virus Entry Mediator Co-Stimulatory Signal Domain Exhibits Exhaustion-Resistant Properties

**DOI:** 10.3390/ijms25168662

**Published:** 2024-08-08

**Authors:** Jun-ichi Nunoya, Nagisa Imuta, Michiaki Masuda

**Affiliations:** Department of Microbiology, Dokkyo Medical University School of Medicine, Tochigi 321-0293, Japanm-masuda@dokkyomed.ac.jp (M.M.)

**Keywords:** chimeric antigen receptor, herpes virus entry mediator, tonic signaling, T cell exhaustion, T cell-mediated immunotherapy

## Abstract

Improving chimeric antigen receptor (CAR)-T cell therapeutic outcomes and expanding its applicability to solid tumors requires further refinement of CAR-T cells. We previously reported that CAR-T cells bearing a herpes virus entry mediator (HVEM)-derived co-stimulatory signal domain (CSSD) (HVEM-CAR-T cells) exhibit superior functions and characteristics. Here, we conducted comparative analyses to evaluate the impact of different CSSDs on CAR-T cell exhaustion. The results indicated that HVEM-CAR-T cells had significantly lower frequencies of exhausted cells and exhibited the highest proliferation rates upon antigenic stimulation. Furthermore, proliferation inhibition by programmed cell death ligand 1 was stronger in CAR-T cells bearing CD28-derived CSSD (CD28-CAR-T cells) whereas it was weaker in HVEM-CAR-T. Additionally, HVEM-CAR-T cells maintained a low exhaustion level even after antigen-dependent proliferation and exhibited potent killing activities, suggesting that HVEM-CAR-T cells might be less prone to early exhaustion. Analysis of CAR localization on the cell surface revealed that CAR formed clusters in CD28-CAR-T cells whereas uniformly distributed in HVEM-CAR-T cells. Analysis of CD3ζ phosphorylation indicated that CAR-dependent tonic signals were strongly sustained in CD28-CAR-T cells whereas they were significantly weaker in HVEM-CAR-T cells. Collectively, these results suggest that the HVEM-derived CSSD is useful for generating CAR-T cells with exhaustion-resistant properties, which could be effective against solid tumors.

## 1. Introduction

Chimeric Antigen Receptor (CAR)-T cell immunotherapy can enhance the acceleration of cancer immunity and has shown remarkable response rates in clinical trials against hematologic malignancies such as B-cell malignancies [1]. Subsequently, it has been approved as a novel therapy in the United States, Europe, Japan, and China [2]. Currently, it is used as a standard treatment for relapsed and refractory cases [3]. CAR is a chimeric molecule with a modular structure consisting of an extracellular antigen recognition domain and an intracellular signaling domain [4]. The signaling domain consists of sequences derived from CD3ζ and a co-stimulatory signaling domain (CSSD). Second-generation CAR-T cells which have one CSSD and third-generation CAR-T cells which have two CSSDs are the mainstream [5]. However, it has become apparent that even in the most studied and clinically applied CD19-targeted CAR-T cell therapy, the relapse rate after complete remission is high. In these relapse cases, it has been observed that CAR-T cells do not persist for long periods in the patient’s body and the expression of CD19 antigen on leukemia cells disappears, which is considered to be the cause of the relapse [6]. Additionally, the anti-tumor effect against solid tumors is known to be very low. One of the contributing factors is thought to be the exhaustion of CAR-T cells [7].

CAR-T cells used in current cancer immunotherapy are also subject to exhaustion in the same way as tumor-infiltrating T cells. For example, mesothelin-specific CAR-T cells have been reported to exhibit increased expression of programmed cell death 1 (PD-1) and lack expression of effector molecules due to exhaustion [8]. Additionally, in patients with chronic lymphocytic leukemia or B-cell lymphoma who did not respond to infusion, CD19-specific CAR-T cells tended to exhibit an exhausted phenotype as revealed by RNA sequencing-mediated transcriptome analysis [9,10]. These reports suggest that CAR-T cell exhaustion is one of the hurdles that need to be overcome. On the other hand, CAR-T cell exhaustion is more complex than that of conventional T cells because CAR itself has signaling domains. A unique mechanism in CAR-T cells is that antigen-independent stimulation via CAR (CAR-dependent tonic signaling) can induce exhaustion during the manufacturing process of CAR-T cells [11,12]. Furthermore, CAR-T cells that should have cytotoxic capabilities against cancer cells fail to suppress tumor progression in the immunosuppressive tumor microenvironment in solid tumors. Since exhaustion and functional inhibition are induced in multiple stages in CAR-T cells, it is currently not possible to fully control CAR-T cell exhaustion.

Among the modular structures that constitute CAR, the CSSD in CAR has been reported to be associated with the function and characteristics of CAR-T cells. When comparing CARs with the representative co-stimulatory molecules CD28 or 4-1BB, CAR-T cells bearing a 4-1BB-derived CSSD (4-1BB-CAR-T cells) have been found to be more resistant to CAR-dependent tonic signaling, making them less prone to early exhaustion. It is also more likely that such cells will maintain high effector function and persist longer in vivo compared to CAR-T cells bearing a CD28-derived CSSD (CD28-CAR-T cells) [11,12]. In the previous study, we have analyzed the utility of CARs with CSSDs derived from herpes virus entry mediator (HVEM), a co-stimulatory molecule belonging to the TNF receptor superfamily, in comparison with the widely used CSSDs such as CD28 or 4-1BB in the production of CAR-T cells. Our results have indicated that among the three compared CSSDs, CAR-T cells bearing an HVEM-derived CSSD (HVEM-CAR-T cells) exhibited the highest effector activity and the least levels of early exhaustion [13]. However, the mechanisms of early exhaustion in CAR-T cells and whether HVEM-CAR-T cells which exhibit low exhaustion can avoid T cell exhaustion and demonstrate superior characteristics even after cognate antigenic stimulation is not fully understood.

We created the HIV Env-targeting CAR-T cells with CSSDs derived from CD28, 4-1BB, and HVEM, as a model system, and evaluated the impact of different CSSDs on CAR-T cell exhaustion. The HVEM-CAR-T cells had significantly lower frequencies of exhausted cells and exhibited the highest proliferation rate upon cognate antigenic stimulation. Proliferation inhibition by programed cell death 1 (PD-L1) was stronger in CD28-CAR-T cells whereas it was weaker in HVEM-CAR-T cells. In addition, HVEM-CAR-T cells maintained a low exhaustion level even after antigen-dependent proliferation and exhibited potent killing activities against HIV-infected CD4 T cells. These data suggested that HVEM-CAR-T cells might be less prone to early exhaustion. CAR formed clusters in CD28-CAR-T cells whereas it was uniformly distributed in HVEM-CAR-T cells. In addition, analysis of CD3ζ phosphorylation indicated that CAR-dependent tonic signals were strongly sustained in CD28-CAR-T cells whereas they were significantly weaker in HVEM-CAR-T cells, consistent with the absence of CAR cluster formation. Therefore, we demonstrate that the HVEM-CAR-T cells exhibit exhaustion-resistant properties.

## 2. Results

### 2.1. HVEM-CAR-T Cells Exhibit Lower Exhaustion Phenotype and Superior Expansion without Antigen Stimulation

We have previously reported that HVEM-CAR-T cells exhibited high functional potency with lower exhaustion status [13]. To confirm their characteristics, we have analyzed the exhaustion status of the CAR-T cells with different CSSDs by using flow cytometry. Consistent with our previous report, the HVEM-CAR-T cells had the lowest frequency of exhausted (PD-1^+^LAG3^+^) cells (Figure 1A,B), suggesting that the HVEM-CAR-T cells exhibited a lower exhaustion phenotype. It has been reported that exhausted T cells possesses a lower expansion rate than healthy T cells [14]. Thus, we also analyzed the expansion of the CAR-T cells during long-term culture after CAR transduction. All the CAR-T cells made in this study expanded without cognate antigen stimulation. The HVEM-CAR-T cells exhibited the highest expansion at days 9 and 16 after CAR transduction (Figure 1C). In addition, the CAR-T cell expansion was inversely correlated with the percentage of exhausted population (Figure 1D). Taken together, these data revealed that HVEM-CAR-T cells exhibited a lower exhaustion phenotype and superior expansion without cognate antigen stimulation.

### 2.2. HVEM-CAR-T Cells Robustly Expand by Cognate Antigen Stimulation Even under Immunosuppression by PD-L1

Since poor proliferation upon antigen stimulation is one of the hallmarks of T cell exhaustion, we next examined CAR-T cell proliferation by cognate antigen stimulation in the presence or absence of PD-L1. We stimulated CAR-T cells using mitomycin C-treated cells and analyzed the CAR-T cell proliferation using flow cytometry (Figure 2A). All CAR-T cells made in this study expanded upon repeated cognate antigen stimulation with HIV Env-expressing cells (Figure 2B, black circles) compared to the stimulation with cells only expressing GFP (Figure 2B, open circles). The proliferation of CAR-T cells upon cognate antigen stimulation with HIV Env-expressing cells was in the order of HVEM-CAR-T cells > 4-1BB-CAR-T cells > CD28-CAR-T cells (Figure 2C,D, left side). Furthermore, CAR-T cells with cells co-expressing HIV Env and PD-L1, CD28-CAR-T cells exhibited significantly stronger growth inhibition when stimulated whereas HVEM-CAR-T cells showed weaker growth inhibition (Figure 2B, grey circles). In addition, HVEM-CAR-T cells exhibited the highest expansion in the presence of PD-L1 among CAR-T cells tested in this study (Figure 2C,D, right side). These data suggested that HVEM-CAR-T cells robustly expanded upon cognate antigen stimulation even under immunosuppression by PD-L1.

### 2.3. HVEM-CAR-T Cells Exhibit Less Exhausted Phenotype after Repeated Cognate Antigen Stimulation

To analyze CAR-T cell characteristics in more detail, we have analyzed expression patterns of inhibitory receptors such as LAG-3 and PD-1 before and after repeated cognate antigen stimulation in the presence or absence of PD-L1. In the absence of PD-L1, the PD-1^+^LAG3^+^ exhausted population of CAR-T cells was robustly elevated after first cognate antigen stimulation (Figure 3A, right side plots in the middle) but declined to almost the same level as their original condition after second stimulation (Figure 3A, right side plots in the bottom). Interestingly, the exhausted population of CAR-T cells declined gradually as a result of repeated cognate antigen stimulation in the absence of PD-L1 (Figure 3A, left side plots in the middle and bottom). Among CAR-T cells tested in this study, CD28-CAR-T cells had the highest frequency of exhausted population but 4-1BB- or HVEM-CAR-T cells had significantly lower frequencies of exhausted population regardless of the presence of PD-L1 (Figure 3B,C). Collectively, these data suggested that unlike CD28-CAR-T cells, HVEM-CAR-T cells exhibited less exhausted phenotype even after repeated cognate antigen stimulation.

### 2.4. HVEM-CAR-T Cells Exhibit Potent Anti-HIV Activity

Next, we examined cytotoxic activity against HIV-infected CD4 T cells. All CAR-T cells made in this study exhibited cytotoxic activity against HIV-infected CD4 T cells. Consistent with our previous report [13], CAR-T cells bearing different CSSDs exhibited different degrees of cytotoxic activity against target cells (Figure 4A). CD28-CAR-T cells exhibited the least cytotoxic activity among the CAR-T cells tested in this study. However, 4-1-BB-CAR-T cells or HVEM-CAR-T cells exhibited higher cytotoxic activity than CD28-CAR-T cells (Figure 4B–D). Especially, HVEM-CAR-T cells exhibited significantly higher cytotoxic activity against HIV-infected CD4 T cells compared to CD28-CAR-T cells or 4-1BB-CAR-T cells at a low effector to target ratio (Figure 4B). These data suggested that HVEM-CAR-T cells exhibited potent anti-HIV activity.

### 2.5. HVEM-CAR-T Cells Form Fewer CAR Microclusters in the Absence of Cognate Antigen Stimulation

Previous study has shown that clustering of the CAR molecules on the cell membrane could induce tonic signaling, causing early exhaustion to CAR-T cells [15]. To reveal the mechanisms of early exhaustion in CAR-T cells bearing different CSSDs, the distribution of CAR molecules on the CAR-T cell surface was analyzed under fluorescent microscopy. We found that CAR molecules were mostly accumulated in the local area on cell membranes of CD28-CAR-T cells (Figure 5A,C, left panels). On the other hand, CAR molecules were equally distributed on the cell membranes of 4-1BB- or HVEM-CAR-T cells (Figure 5C, middle and right panels). The numbers of CAR-T cells with CAR puncta were significantly higher in CD28-CAR-T cells than in 4-1BB- or HVEM-CAR-T cells (Figure 5B,D). In addition, HVEM-CAR-T cells had the lowest CAR puncta formation among CAR-T cells tested in this study (Figure 5D). Interestingly, these CAR microclusters in CD28-CAR-T cells were formed at the early stage of culture (Figure 5A, day 9) and lasted long-term (Figure 5C, day 24). Collectively, these data suggested that HVEM-CAR-T cells had fewer CAR microclusters in the absence of cognate antigen stimulation.

### 2.6. HVEM-CAR-T Cells Exhibit Lower CAR-Mediated Tonic Signaling in the Absence of Cognate Antigen Stimulation

To examine whether tonic signaling has been associated with CAR microcluster formation, the levels of phosphorylation in tyrosine 142 in CD3ζ of the CAR were analyzed by flow cytometry. As shown in our previous report [15], CD3ζ phosphorylation could not be detected in non-transduced populations (Appendix A), demonstrating that CD3ζ phosphorylation specifically occurred in the CAR-T cells (Figure 6A). The level of CD3ζ phosphorylation was higher in the CD28-CAR-T cells than in the 4-1BB- or HVEM-CAR-T cells at days 9 and 16. Thus, these data indicated that CAR-mediated tonic signaling was specifically induced and lasted for a long time in the CD28-CAR-T cells (Figure 6B) but not in the 4-1BB- or HVEM-CAR-T cells. Taken together, HVEM-CAR-T cells exhibited lower CAR-mediated tonic signaling in the absence of cognate antigen stimulation.

## 3. Discussion

CAR-T cell-based immunotherapy is expected to become a valuable therapeutic intervention against diseases associated with impaired cellular immunity such as malignant tumors. While CAR-T cells have achieved very high therapeutic outcomes in hematological malignancies, it has been reported that solid tumors were unresponsive to CAR-T cells [16]. One known contributing factor to this issue is CAR-T cell exhaustion, which has not been adequately overcome yet. Therefore, further improvement of CAR-T cells is necessary for practical application. To this end, newer generations of CARs are being developed by modifying components of the CAR modular structure, such as the CSSD. For instance, our group recently reported that more effective CAR-T cells can be produced using an HVEM-derived CSSD, instead of the commonly used CD28- or 4-1BB-derived CSSDs [13]. However, it remains unclear whether HVEM-CAR-T cells can avoid T cell exhaustion and exhibit superior characteristics after repeated antigen stimulation. In order to investigate this issue, we constructed vectors for expressing CARs bearing different CSSDs. Analysis of the primary T cells transduced with these vectors demonstrated that the level of CAR-T cell exhaustion and the strength of CAR-dependent tonic signaling varied depending on the type of CSSD used in the CAR. In addition, we discovered a positive correlation between the degree of CAR clustering on the cell surface, the strength of CAR-dependent tonic signaling, and the level of CAR-T cell exhaustion. This suggested that the clustering of CARs on the cell surface transmits tonic signals, leading to early exhaustion of the CAR-T cell. Additionally, upon specific antigenic stimulation, HVEM-CAR-T cells with a lower early exhaustion level responded and proliferated well. Even after proliferation, these HVEM-CAR-T cells maintained low exhaustion levels and exhibited superior characteristics. These results suggest that CARs with an HVEM-derived CSSD can confer exhaustion-resistant properties on CAR-T cells, potentially improving therapeutic efficacy.

CAR is a chimeric molecule with a modular structure including an intracellular signaling domain. Non-specific signal transduction called CAR-mediated tonic signaling is known to occur without cognate antigen binding [17], causing exhaustion and dysfunction in CAR-T cells [18]. CAR-mediated tonic signaling is thought to be due to their modular structural components and architecture in the CARs [19]. For example, when the CH_2_-CH_3_ region of IgG1 is used in a spacer between scFv and the transmembrane region of the CAR, stronger CAR-mediated tonic signals occur than when only CH3 is used [20]. Slight structural differences in the framework region within scFv in the CAR are also able to induce antigen-independent clustering of the CAR and tonic signaling [21]. Modulating V_H_-V_L_ and V_L_-V_H_ orientations in the scFv region of the CAR has also affected intensity of CAR-mediated tonic signaling and the percentage of exhausted cells [15]. In addition, CSSD in the CAR also affected the strength of CAR-mediated tonic signaling. Consistent with previous reports [11,12,13,22], this study has shown that 4-1BB-CAR-T cells and HVEM-CAR-T cells are more resistant to CAR-mediated tonic signaling than CD28-CAR-T cells. Both 4-1BB and HVEM belong to the TNF receptor superfamily, suggesting the advantage of using TNF receptor superfamily CSSDs in the creation of effective CARs [23].

Early exhaustion of CAR-T cells can significantly limit their therapeutic efficacy and negatively impact patient outcomes [16]. In this study, we analyzed the effects of early exhaustion of CAR-T cells on their proliferation in response to antigenic stimulation and the exhaustion status of proliferated CAR-T cells. Our results showed that CD28-CAR-T cells with a high degree of early exhaustion exhibited a significant increase in the proportion of exhausted cells associated with lower proliferation rates upon antigenic stimulation. In contrast, 4-1BB-CAR-T cells and HVEM-CAR-T cells had a significantly lower proportion of exhausted cells and exhibited higher proliferation rates after antigenic stimulation. In the presence of PD-L1, highly exhausted CD28-CAR-T cells proliferated in response to the initial antigen stimulation but showed minimal proliferation thereafter. On the other hand, 4-1BB-CAR-T cells and HVEM-CAR-T cells responded to repeated antigen stimulation and maintained a consistent proliferative capacity even in the presence of PD-L1 (Figure 2 and Figure 3). These findings suggest that the early exhaustion status of CAR-T cells affects their proliferative response to antigenic stimulation. In addition, it has been reported that CAR-T cells with different CSSDs exhibit variations in energy metabolism and memory phenotypes [13,22]. Given that intracellular signaling can lead to changes in cell characteristics and functions, CSSDs are considered major determinants in the induction of early exhaustion in CAR-T cells. Also, HVEM-CAR-T cells, which are less prone to early exhaustion, exhibited high functional potency, especially in a significant killing activity on target cells even at low E:T ratios (Figure 4). Although various factors are involved in the characteristics and functions of CAR-T cells, making it difficult to elucidate all of them, CARs with an HVEM-derived CSSD may confer exhaustion resistance to CAR-T cells, potentially improving therapeutic outcomes.

Typically, repeated stimulation is thought to induce more exhausted T cells. As seen in Figure 3, the phenomenon where the population of exhausted cells decreases to the original level after the second stimulation is not fully understood. In the results presented here, cells one week after each stimulation were used for analysis. Therefore, in the results after second stimulation, it is possible that the exhausted cells died due to apoptosis during the week before the analysis, making it appear that the population of exhausted cells decreased.

Also, we used CAR-T cells targeting the HIV Env protein as a model system in this study. As mentioned earlier, CAR has a modular structure, and changes in the structure of the CAR have been reported to affect the characteristics and functions of CAR-T cells [15,19,20,21]. CAR-T cells with exhaustion-resistant properties are considered particularly useful in the development of CAR-T cells targeting solid tumors such as pancreatic cancer. Therefore, it is important to verify the utility of the HVEM-derived CSSD in CAR-T cells targeting solid tumors in the near future.

Avoiding exhaustion is one of the useful strategies to control the characteristics and functions of CAR-T cells for obtaining maximal therapeutic efficacy [24]. As mentioned above, switching V_H_ and V_L_ orientation or changing the CSSD in the modular structure of the CAR can minimize CAR-T cell exhaustion [13,15]. On the other hand, the expression of inhibitory receptors induced by sustained antigenic stimulation induces inhibitory signals, which partly leads to functional inhibition of CAR-T cells. From this point of view, the use of CAR-T cells in combination with immune checkpoint inhibitors (ICIs) or secreting ICIs from CAR-T cells has been tried and shown some efficacy [8,25,26,27,28]. Another strategy is focused on transcription factors for avoiding exhaustion in CAR-T cells. Overexpression of the transcription factor c-Jun in CAR-T cells has been reported to induce enhanced proliferation and function, resist exhaustion, suppress differentiation, and exhibit enhanced anti-tumor activity [29]. The transcription factor NR4A has been shown to limit CAR-T cell functions and deletion of NR4A reduces exhaustion and confers potent anti-tumor activity and viability [30]. Deletion of the PR domain zinc finger protein 1 (PRDM1) gene at the manufacturing stage has also been able to reduce exhaustion and to produce CAR-T cells with enhanced viability [31]. Thus, by controlling the degree of CAR-T cell exhaustion at the manufacturing stage, next-generation CAR-T cells that exhibit better properties and higher anti-tumor efficacy than ever before are being developed for future immunotherapy.

In summary, this study demonstrated that the CSSD in a CAR is a crucial determinant for early exhaustion in CAR-T cells, indicating that the CSSD in a CAR is important for designing CAR-T cells with exhaustion-resistant properties. Moreover, the HVEM-CAR-T cells may be a promising candidate for generating effective CAR-T cells against solid tumors such as pancreatic cancer.

## 4. Materials and Methods

### 4.1. Cell Culture

Minimum essential medium (MEM) and Dulbecco’s modified Eagle medium (DMEM) (Thermo Fisher Scientific, Waltham, MA, USA) were supplemented with 10% FBS (Thermo Fisher Scientific), 2 mM glutamine (Thermo Fisher Scientific), 10 U/mL penicillin, and 10 μg/mL streptomycin (Thermo Fisher Scientific) and named as M10 and D10, respectively. Chinese hamster ovary (CHO) cells expressing GFP (CHO-GFP) or both HIV-1 Env and GFP (CHO-Env-GFP) were maintained in M10 supplemented with non-essential amino acids (NEAA) containing 5 mg/mL blastcidin (BSD) [13]. To prepare target cells for stimulation of CAR-T cells, sub-confluent CHO-GFP or CHO-Env-GFP cells on 150 mm culture dish were cultured in the presence of 10 μg/mL mitomycin C (Tokyo Chemical Industry, Tokyo, Japan) for 3 h. Cells washed with PBS were harvested with a cell scraper (Corning, Corning, NY, USA). Single cell suspension was obtained through 70 μm nylon cell strainer (Corning). The 293FT cells were cultured in D10. Human peripheral blood mononuclear cells (PBMCs) from healthy donors were prepared by Ficoll-Paque (GE Healthcare Life Sciences, Pittsburgh, PA, USA) density gradient and cultured in AIM-V (Thermo Fisher Scientific) supplemented with 5% FBS, 10 mM HEPES (complete AIM-V) overnight to remove adherent monocytes. The monocyte-depleted PBMCs were used in transduction experiments. All of the cells were grown at 37 °C with 5% CO_2_.

### 4.2. Vector Constructions

sCD4-CAR with different CSSD expressing lentiviral vector plasmids were constructed previously [13]. The *Xba*I/*Eco*RI-digested sCD4 cDNA fragment were prepared as previously described [13] (fragment A). To remove the stop codon from CAR encoding cDNA, the cDNA fragment encoding the CAR signaling domain was amplified by PCR, as previously described [15], using the following primers: Signal-F (5′-gag ccc ccc tgg tag tag ccc ctc ag-3′) and CD3z-R (5′-tta tat agg ggc gga tcc gcg agg tgg cag-3′). Amplified fragments were digested with *Eco*RI/*Bam*HI, and purified for ligation (fragment B). The cDNA fragments encoding GS linker (GGGGSGGGGSGGGGS) and emGFP were synthesized and cloned in pUC57 plasmid by Genscript. The cDNA fragment C was prepared by digesting the plasmid with *Bam*HI and *Bsi*WI. To construct the lentiviral vectors expressing CAR-emGFP fusion protein, the cDNA fragments A, B, and C were ligated into *Xba*I/*Bam*HI digested pTK643-EF1a [13].

### 4.3. Preparation of the Vector Virus and HIV-1

The vector lentivirus was prepared as described previously [13] with some modifications. Briefly, 293FT cells were cultured at 80 to 90% confluency on the 150 mm dish (IWAKI, Shizuoka, Japan) coated with poly-L-lysine (Nacalai Tesque, Kyoto, Japan). The culture medium was replaced with D10 containing 25 μM chloroquine (SIGMA, Darmstadt, Germany). Then, the cells were co-transfected with the lentiviral vector plasmid (39 μg), ΔNRF (26 μg), and pMD.G (13 μg) by using the polyethyleneimine (PEI) “MAX” reagent (Polysciences, Warrington, PA, USA) at DNA-to-PEI ratio of 2:1. After transfection, the culture medium was replaced with D10 containing 5 mM sodium butylate (WAKO, Osaka, Japan) and 10 μM forskolin (Tokyo Chemical Industry, Tokyo, Japan). The culture supernatants were harvested at 48 h after transfection, cleared by centrifugation and 0.45 μm filtration (Millipore, Burlington, MA, USA). For preparing the concentrated vector virus, the filtered samples were subjected to high-speed centrifugation at 42,200× *g* for 3 h using a Himac CR21N (Hitachi Koki, Tokyo, Japan). Titers of the concentrated vector were measured on HeLa cells based on the percentage of GFP-positive cells by flow cytometry. For propagation of HIV-1, 293FT cells were transfected with pNL4-3 [32] obtained from Dr. Malcom Martin (through the NIH AIDS Reagent Program) by using the PEI “MAX” reagent as described above. The viral titers were measured on U337-MAGI-X4 cells [33] obtained from Dr. Michael Emerman (through the NIH AIDS Reagent Program) according to the protocol provided by the AIDS Reagent Program. Propagation of HIV-1 was performed in a biosafety level (BSL) 3 facility.

### 4.4. Transduction with Lentiviral Vectors

Human primary CD4^+^ and CD8^+^ T cells were isolated from monocyte-depleted PBMCs by using a FACS Aria II cell sorter (BD Biosciences, Franklin Lakes, NJ, USA) with anti-human CD3-APC (RRID: AB_314048), CD4-PE (RRID: AB_571955), and CD8-PE/Cy7 (RRID: AB_314118) antibodies (all from Biolegend, San Diego, CA, USA), and a purity of more than 95% was routinely achieved. The purified CD8^+^ T cells were activated with anti-CD3/CD28 beads (Thermo Fisher Scientific) at bead-to-cell ratio of 3:1 in complete AIM-V supplemented with 40 U/mL recombinant human IL-2 (obtained from Dr. Maurice Gately through NIH AIDS Reagent Program). On the next day, activated CD8^+^ T cells were transduced with concentrated recombinant lentivirus by using plates coated with Retronectin (TAKARA, Shiga, Japan) according to the manufacturer’s instructions. The cells transduced with the vector expressing only GFP were used as a control. After incubation for 48 h, the supernatant was replaced with complete AIM-V supplemented with 300 U/mL of recombinant human IL-2. The anti-CD3/CD28 beads were removed at 48 h after changing the media. Thereafter, the cells were cultured for 16–18 days by replenishing fresh media containing IL-2 (300 U/mL) every 2 to 3 days.

### 4.5. Flow Cytometric Analysis

For flow cytometry, the centrifuged cells were resuspended in the fluorescence-activated cell sorter (FACS) buffer (PBS containing 2% FBS and 0.02% sodium azide) containing the antibodies and incubated on ice for 30 min unless otherwise indicated. After washing with ice-cold FACS buffer, the cells were fixed with 1% paraformaldehyde/PBS and analyzed by using a flow cytometer (FACS Calibur or LSRFortessa X-20; BD Biosciences). The data were analyzed by FlowJo 10.9.0 (BD Biosciences). The following antibodies used for flow cytometry were obtained from Biolegend unless indicated otherwise. The CAR expression on the vector-transduced primary CD8^+^ T cells was analyzed with biotinylated anti-c-myc tag (RRID: AB_2565090) antibody followed by streptavidin-PE (TONBO biosciences, San Diego, CA, USA). For evaluating exhaustion of the CAR-T cells, flow cytometry was performed with anti-human PD-1-APC (RRID: AB_940475) and anti-human LAG-3-PE/Cy7 (RRID: AB_2629753) antibodies. For detecting CD3ζ phosphorylation in CAR-T cells, the vector-transduced primary CD8^+^ T cells were stained with biotinylated anti-c-myc tag antibody followed by streptavidin-PE. Stained cells were fixed at 37 °C for 10 min in Fix Buffer I (BD) and permeabilized at 4 °C for 30 min in Perm Buffer III (BD). Permeabilized cells were incubated with anti-CD247 (pY142)-PE (RRID: AB_647237) antibody (BD) at room temperature for 30 min.

### 4.6. Microscopic Analysis of CAR Cluster Formation

The CAR-emGFP fusion gene-transduced primary CD8^+^ T cells were sorted at days 9 and 24 based on emGFP expression using a FACS Aria II and seeded on a flat-bottomed 96-well cell culture plate. The sorted cells were observed using a BZ-9000 fluorescent microscopy system (KEYENCE, Osaka, Japan) with a 20× nonoil objective lens. The phase contrast and fluorescent images were taken and analyzed by using ImageJ 1.54 (National Institute of Health, MD) [15]. Briefly, the individual cell was identified from phase contrast and fluorescent images. Manual analysis was performed on a hundred randomly picked cells from each image by applying a threshold, and using the “analyze particles” ImageJ function.

### 4.7. Stimulation of the CAR-T Cells with the Cells Expressing the Cognate Antigen

The CAR-T cells (0.5 × 10^6^) at day 18 were stimulated with mitomycin C-treated CHO-GFP, CHO-Env-GFP, or CHO-Env-PDL1-GFP cells (1 × 10^6^) in 2 mL of complete AIM-V supplemented with 300 U/mL IL-2. The cells were cultured for 7 days by replacing half of the culture media with the fresh media containing 300 U/mL IL-2 every 2 to 3 days. On day 7, the cells in half of the culture were stimulated again with mitomycin C-treated CHO-GFP, CHO-Env-GFP, or CHO-Env-PDL1-GFP cells (1 × 10^6^) in 2 mL of complete AIM-V supplemented with 300 U/mL IL-2 and cultured for 7 days. Flow cytometric analyses to determine CAR-T (CD3^+^GFP^+^) cell expansion were performed on days 7 and 14 after the initial stimulation.

### 4.8. Co-Culture for Anti-HIV-1 Activity of the CAR-T Cells

CD4^+^ T cells, isolated from the same donor of CD8^+^ T cells which were used for generating CAR-T cells, were activated with anti-CD3/CD28 beads at a bead-to-cell ratio of 3:1 in complete AIM-V supplemented with 40 U/mL IL-2. The anti-CD3/CD28 beads were removed at day 3 and cultured for 9 days with replenishing fresh media containing 40 U/mL IL-2 every 2–3 days. Activated CD4^+^ T cells (1 × 10^5^) were exposed with HIV-1 (NL4-3 strain) at MOI = 0.1 for 3 h. After removal of unbound virus, the virus-infected cells were cultured overnight. HIV-1-infected CD4^+^ T cells were co-cultured for 3 days with CAR-T cells at E:T ratios of 1:1, 0.1:1, and 0.01:1. Cells were harvested and stained with anti-human CD8-PE/Cy7 (Biolegend) as described above. Intracellular HIV-1 p24 expression was detected using an Intracellular Fixation and Permeabilization Buffer Set (eBioscience) with anti-HIV-1 p24 antibody KC57-RD1 (Beckman Coulter, RRID: AB_6604667) according to the manufacturer’s instructions.

### 4.9. Statistics

Statistical analysis was described previously [13,15]. A two-tailed Mann–Whitney U test and an unpaired one-way or two-way analysis of variance (ANOVA) with Bonferroni or Tukey multiple comparison tests were performed using GraphPad Prism 10.1.2 for Windows (GraphPad Software, San Diego, CA, USA). A *p* value of <0.05 was considered statistically significant.

## Figures and Tables

**Figure 1 ijms-25-08662-f001:**
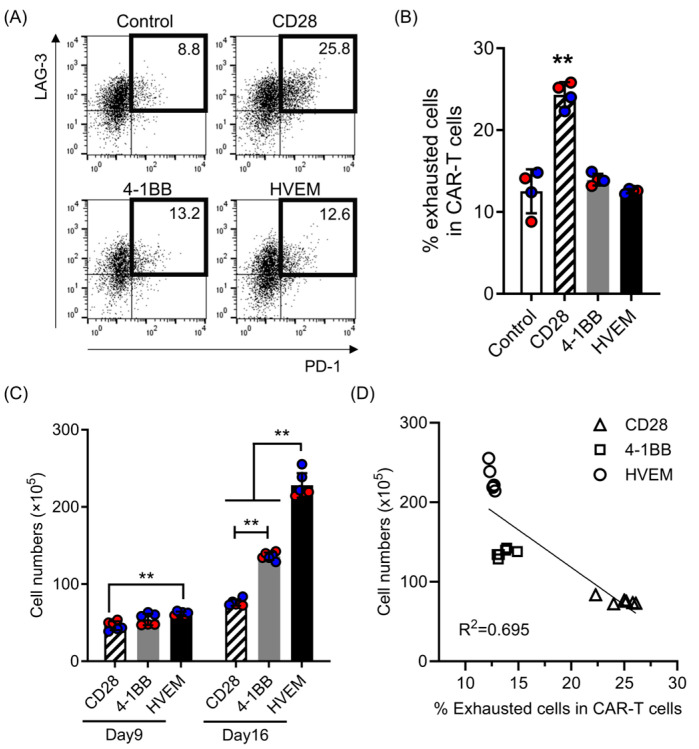
HVEM-CAR-T cells exhibit lower exhaustion phenotype and superior expansion without cognate antigen stimulation. (**A**) Expression of LAG-3 (y-axis) and PD-1 (x-axis) on each CAR-T cell was analyzed at day 16 post CAR transduction by flow cytometry. Dot plots show representative data from each CAR-T cell population. Numbers in the dot plot show the percentage of LAG-3+/PD-1+ (exhausted) population in the CAR-T cells. (**B**) The bar graph shows percentages of the exhausted cells (y-axis) among the CAR-T cells at day 16 of CAR transduction. Control shows T cells transduced only with GFP. Each bar shows the mean with standard deviations (SDs) with each data point as an open circle. The results from two separate experiments using primary CD8^+^ T cells from two different donors (*n* = 4) are shown. ** *p* < 0.05 (one-way ANOVA; Tukey post hoc test). (**C**) The bar graph shows CAR-T cell expansion (y-axis) during culture at day 9 and day 16 of CAR transduction. Each bar shows the mean with SDs with each data point as a filled circle. The results from two separate experiments using primary CD8^+^ T cells from two different donors (*n* = 6) are shown in red- or blue-filled circles. ** *p* < 0.05 (two-way ANOVA; Bonferroni post hoc test). (**D**) Linear regression analysis between the percentages of expanded CAR-T cell numbers (y-axis) and the percentages of exhausted cells among CAR-T cells (x-axis) at day 16 of the culture are shown.

**Figure 2 ijms-25-08662-f002:**
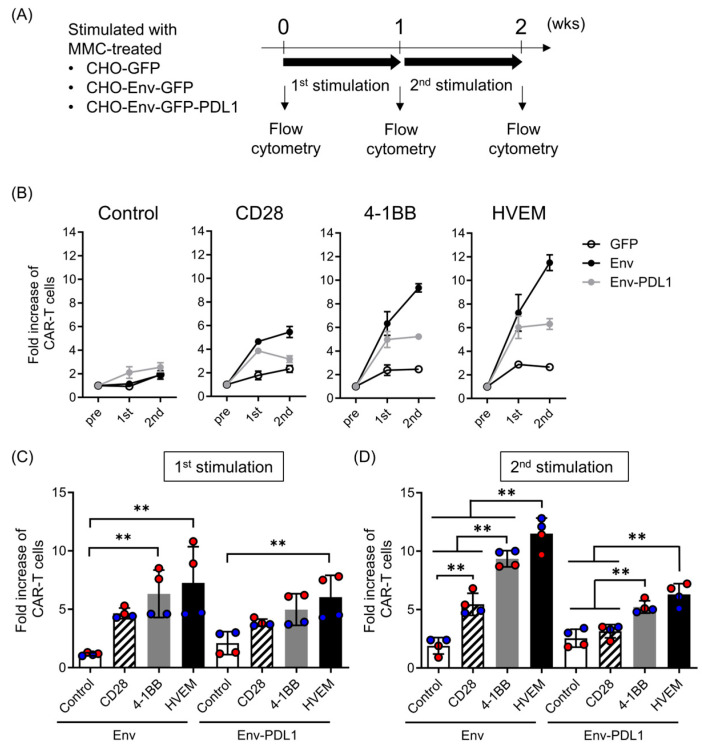
HVEM-CAR-T cells robustly expand upon cognate antigen stimulation even under immunosuppression by PD-L1. (**A**) Schematic image for determining CAR-T cell expansion upon cognate antigen stimulation in the presence or absence of PD-L1. CAR-T cells bearing different CSSDs were stimulated with mitomycin C-treated cells (listed in (**A**), left) and the CAR-T cell proliferation was analyzed using flow cytometry. (**B**) The graphs show fold expansion of CAR-T cells after first (1st) and second (2nd) stimulation with mitomycin C-treated cells only expressing GFP (labeled as GFP, open circles), GFP and Env (labeled as Env, filled black circles), or GFP and Env and PD-L1 (labeled as Env-PDL1, filled grey circles). (**C**,**D**) The bar graph shows fold expansion of CAR-T cells (y-axis) after first (**C**) and second (**D**) stimulation with mitomycin C-treated CHO-Env-GFP (left) or CHO-Env-GFP-PD-L1 (right). Each bar shows the mean with SDs with each data point as a filled circle. The results from two separate experiments using primary CD8 T cells from two different donors (*n* = 4) are shown in red- or blue-filled circles. ** *p* < 0.05 (one-way ANOVA; Tukey post hoc test).

**Figure 3 ijms-25-08662-f003:**
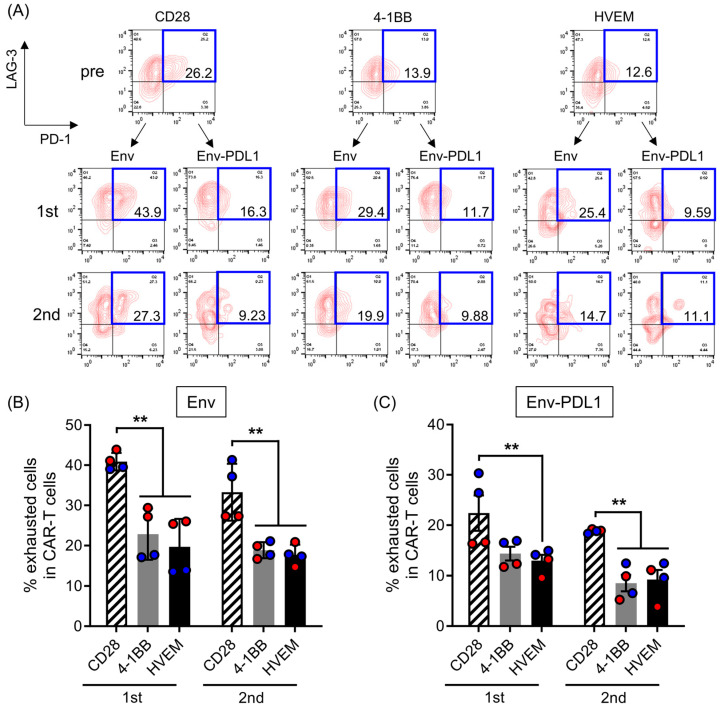
HVEM-CAR-T cells exhibit less exhausted phenotype even after repeated cognate antigen stimulation. (**A**) Representative counter plots show LAG-3 (y-axis) and PD-1 (x-axis) expression in CAR-T cells bearing different CSSDs before (pre), after first (1st), and after second (2nd) stimulation using mitomycin C-treated CHO-Env-GFP (labeled as Env) or CHO-Env-GFP-PD-L1 cells (labeled as Env-PDL1). Numbers in the dot plot show the percentage of LAG-3+/PD-1+ (exhausted) population in the CAR-T cells. (**B**,**C**) The bar graph shows percentages of the exhausted cells (y-axis) among the CAR-T cells after first (left) and second (right) stimulation with mitomycin C-treated CHO-Env-GFP (**B**) or CHO-Env-GFP-PD-L1 cells (**C**). Each bar shows the mean with SDs with each data point as a filled circle. The results from two separate experiments using primary CD8 T cells from two different donors (*n* = 4) are shown in red- or blue-filled circles. ** *p* < 0.05 (one-way ANOVA; Tukey post hoc test).

**Figure 4 ijms-25-08662-f004:**
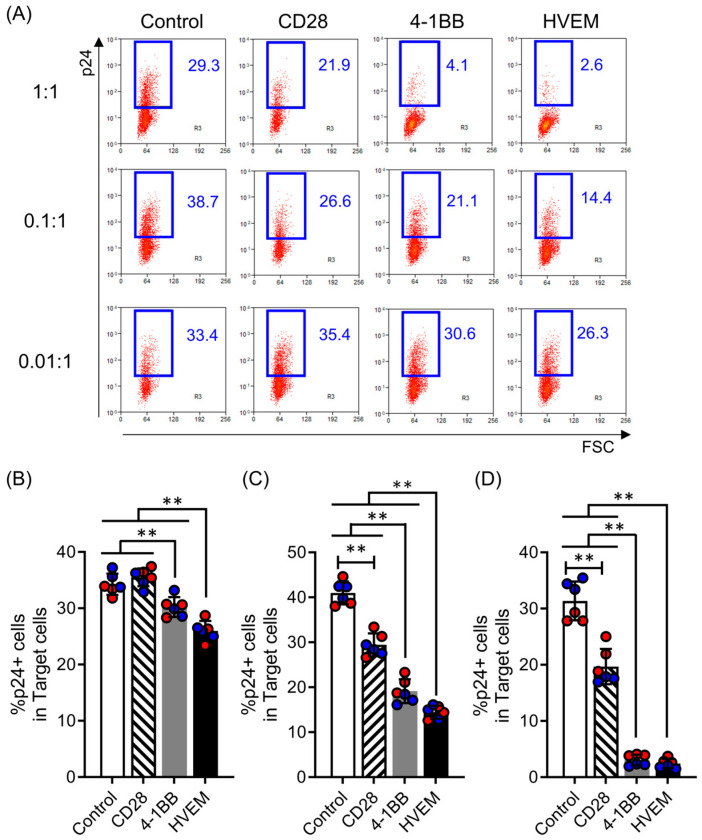
HVEM-CAR-T cells exhibit potent anti-HIV activity. (**A**) HIV-1-infected CD4^+^ cells were co-cultured with the control cells or CAR-T cells bearing different CSSDs at E:T ratios of 1:1, 0.1:1, and 0.01:1. FSC (x-axis) and HIV p24 expression (y-axis) were analyzed after 3 days of co-culture by flow cytometry. Dot plots show representative data from each type of CAR-T cell and E:T ratio. Numbers in the dot plot show the percentage of HIV p24^+^ (HIV-infected) cells in CD8^−^ population. (**B**–**D**) The bar graph shows percentage of HIV p24^+^ cells in CD8^−^ population in the co-culture at E:T ratios of 0.01:1 (**B**), 0.1:1 (**C**) and 1:1 (**D**). Control shows T cells transduced only with GFP. Each bar shows the mean with SDs with each data point as a filled circle. The results from two separate experiments using primary CD8^+^ T cells from two different donors (*n* = 6) are shown in red- or blue-filled circles. ** *p* < 0.05 (one-way ANOVA; Tukey post hoc test).

**Figure 5 ijms-25-08662-f005:**
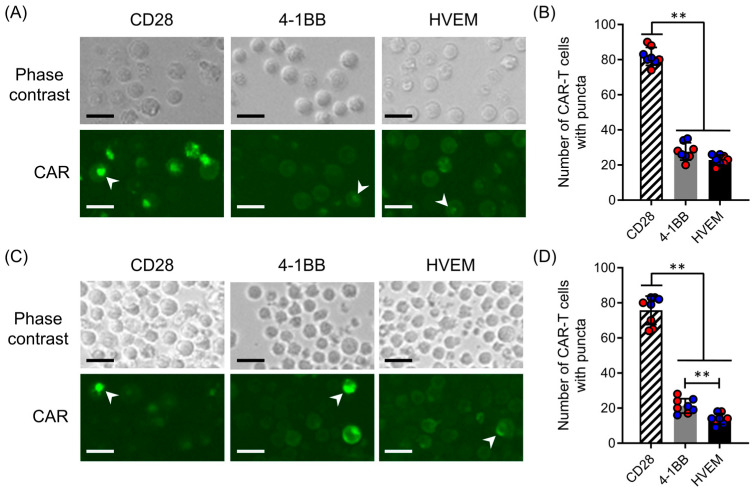
HVEM-CAR-T cells form fewer CAR microclusters in the absence of cognate antigen stimulation. Imaging analysis of cell surface CAR distributions on sorted CAR-T cells bearing different CSSDs at day 9 (**A**,**B**) and day 24 (**C**,**D**) after CAR transduction. The representative phase contrast (upper) and CAR (lower) images at day 9 (**A**) and day 24 (**C**) are shown. White arrowhead in each image shows the representative morphology of a CAR microcluster. Scale bars = 25 μm. Bar graph shows the number of CAR-T cells with CAR puncta at day 9 (**B**) and day 24 (**D**). Data of 100 randomly chosen CAR-T cells in each image are shown. Each bar shows the mean with standard deviations with each data point as a filled circle. The results from two separate experiments with primary CD8^+^ T cells from two different donors (*n* = 8) are shown in red- or blue-filled circles. ** *p* < 0.05 (one-way ANOVA; Tukey post hoc test).

**Figure 6 ijms-25-08662-f006:**
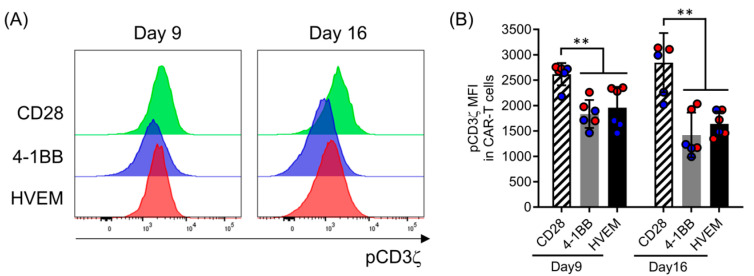
HVEM-CAR-T cells exhibit lower CAR-mediated tonic signaling in the absence of cognate antigen stimulation. (**A**) Panels show a typical phosphorylated CD3ζ histogram of the successfully transduced GFP^+^ populations of the CAR-T cells bearing different CSSDs at day 9 and day 16 analyzed by flow cytometry. (**B**) Bar graphs show the MFI of phosphorylated CD3ζ in the GFP^+^ populations of control T cells or CAR-T cells bearing different CSSDs at day 9 (left) and day 16 (right). Each bar shows the mean with SDs with each data point as a filled circle. The results from two separate experiments with primary CD8^+^ T cells from two different donors (*n* = 6) are shown in red- or blue-filled circles. ** *p* < 0.05 (two-way ANOVA; Bonferroni post hoc test).

## Data Availability

Dataset available upon reasonable request from the authors.

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
