# Peer review of "Chimeric Antigen Receptor T Cell Bearing Herpes Virus Entry Mediator Co-Stimulatory Signal Domain Exhibits Exhaustion-Resistant Properties"

_ijms, 2024, doi:10.3390/ijms25168662_

Round 1

Reviewer 1 Report

Comments and Suggestions for Authors

Altogether a well written paper introducing the HVEM-CAR-T cells as potential therapeutic application.

In the discussion you could potentially include a small paragraph really putting forward the use of HVEM in comparison to 4-1BB as you put the CSSD in focus and you were able to see clear differences in the results especially in micro-cluster formation on day 24 as well as cell numbers and fold increase. 

Otherwise it is a well performed study, and would just need some revising of the text for readability.

Comments on the Quality of English Language

Sentence formation needs some re-do. E.g. 98 . The all CAR-T cells made... -> All Car T cells. 

Just needs a readthrough and some changes, nothing severe.

Reviewer 2 Report

Comments and Suggestions for Authors

The authors describe the functionalities of a relatively novel type of CAR-T cells, which employ a herpes virus entry mediator (HVEM)-derived co-stimulatory signal domain (CSSD), in comparison with CAR-T cells transduced with vectors using CD28- and 4-1BB derived CSSDs. They show that these cells exhibit low frequencies of exhausted cells and high proliferation rates upon antigenic stimulation comparing with other cells used. Their proliferation cannot be inhibited with PD-L1 as efficiently as that of CD28-CAR-Ts. These cells maintained low exhaustion phenotype even after antigen stimulation, and show very potent killing activity. Microscopy analysis shows that the chimeric TCRs in the cells do not cluster and remain uniformly distributed. CAR-dependent tonic signals, analysed using CD3zeta phosphorylation in FACS assay, are low in comparison with CD28 CAR-T cells, which may contribute to slow exhaustion or the novel cell type.

The present manuscript is a valuable contribution to the body of knowledge and potential alternatives in designing CAR-T cells, which are promising but challenging therapeutics. The text is clearly written, the inference is clear and the conclusions well drawn. At points (please see the remarks below), the methods should be outlined in more detail. Please find below a list of remarks, which I hope will be helpful.   

Line 19: CD3zeta, zeta with the Greek symbol

Line 48: “CAR-T cells tended to exhibit an exhausted phenotype” – I propose to specify here what this phenotype is and how it was measured: is it lack of expression of effector molecules, as mentioned earlier?

Line 62: “It is also more likely to maintain high effector function” – it is also more likely that such cells will maintain high effector function

Line 66: “in addition to the widely used CSSDs” – in comparison with the widely used CSSDs

Line 75: “from CD28, 4-1BB, and HVEM as a model system, and evaluate the impact…”: from CD28, 4-74 1BB, and HVEM, as a model system, and evaluated the impact…

Line 82: “exhaustion. Mechanistically, CAR formed clusters in CD28-CAR-T cells whereas uniformly distributed in HVEM-CAR-T cells” – CARs formed clusters in CD28-CAR-T cells while they were uniformly distributed…”

Line 84: „CD3z“ – z should be zeta (Greek letter)

Line 89: “HVEM-CAR-T cell exhibits lower exhaustion phenotype” - HVEM-CAR-T cells exhibit lower exhaustion phenotype. It is more usual to keep the cell type in plural, so please correct throughout the manuscript

Figure 2 C and D: The presentation with single data points is very good, but it is not possible to see all data points in the black bars. The measurements from 2 individuals could be in different markers.

Line 173: “Representative dot plots“ – these are contour plots

Line 197: “(open circle)“ – there are no open circles in 4A

Line 200: show the percentage

Line 226: something is wrong with micrometer symbol

Line 238: “phosphorylation was specifically occurred” – phosphorylation specifically occurred

Lines 248 and 251: something is wrong with zeta symbols

Line 254: Bonferroni

Line 269: conducted the vectors: constructed?

Line 284: known to be occurred without cognate antigen binding – known to occur

Line 331: “are able to avoid CAR-T cell exhaustion” – are able to minimize CAR T cell exhaustion

Line 357 and 363: something wrong with the symbol for micro (please correct throughout the text)

Line 360: non-essential amino acids

Line 365: 70 micrometer

Line 366: where were Jurkat E6.1 cells used?

Line 400: “centrifugation at 18,000 rpm“ – should be in g units

Line 403: Format of the inline citation 32

Line 446: The Image J-based method for evaluation of puncta requires a more precise description. Also reference 15 does not give more details than are cited here.

Line 448: all antibodies should be mentioned with RRIDs

Reviewer 3 Report

Comments and Suggestions for Authors

In the current manuscript the authors evaluate the impact of co-stimulatory signal domain (CSSD) on CAR-T cell exhaustion. Conducting in vitro analyses using HIV Env-targeting CAR-T cells engineered with CD28, 4-1BB, and HVEM CSSDs, the authors show that HVEM-CAR-T cells have lower frequencies of exhausted cells, higher proliferative capacity, and increased cytotoxic ability upon cognate antigenic stimulation.

The rationale for the study is sound and the study design is appropriate and presented clearly. The data presented build logically off the authors’ previous manuscript and are largely supportive of the authors’ conclusions. I have one major comment that needs to be addressed before publication is recommended.

Major comments:

1. While the experimental data in Figure 6 is suggestive that tonic signaling is reduced in HVEM-CAR T cells compared to CD28-CAR T cells, it is critical to understand whether the proportion of phosphorylated CD3 zeta among the total CD3 zeta protein differs between the two populations. I suggest that the authors conduct Western Blot analyses of transduced, unstimulated cells at the time points shown in Figure 6 (i.e., day 9 and 16).

Minor comments:

1. line 34: since CD3z is used predominately throughout the manuscript, CD3z should be changed to CD3z for consistency.

2. line 98: “The all” should be corrected to “All the”.

3. line 127: remove the extra spaces in the word analyzed.

4. line 164: correct “tasted” to “tested”.

Reviewer 4 Report

Comments and Suggestions for Authors

Jun-ichi Nunoya et al. created the HIV Env-targeting CAR-T cells with CSSDs derived from CD28, 4-1BB, and HVEM as a model system, and evaluate the impact of different CSSDs on CAR- T cell exhaustion.Jun-ichi Nunoya et al. demonstrate that the HVEM-CAR-T cells exhibit exhaustion-resistant properties.I think the quality of this article is not high, the manuscript has problems such as less experimental data and lack of experimental control.

1. In the Figure.1C, the authors chose days 9 and 16. Why did they choose this time? Can the authors provide all the data from Day 1 to Day 20?

2. Figure.1A Why do they and Figure.3C choose different colors?(One gray, one in color)

3. The first stimulation and second stimulation related confidence need to be annotated in Figure 2C and 2D, rather than just illustrated in the figure legends. This increases the readability of the article.

4. The authors need to explain the experimental results in Figure 3, such as why such a result as "In the absence of PD-L1, the PD-1+LAG3+ exhausted population of CAR-T cells was robustly elevated after first cognate antigen stimulation but declined to almost same level of original condition after second stimulation" appears. The authors are necessary to add experiments for interpretation.

5. .I am not sure whether the authors can reliably conclude that the "HVEM-CAR-T cell exhibits potent anti-HIV activity" works by using only one experiment (Figure4).

6. There is no control group in Figure 5, and the authors need to mark the morphology of the "CAR microcluster", as well as the morphology of the control group.

7. In Figure 6, the authors need to supplement the WB experiments to verify it, not only using flow cytometry experiments. In addition, why did the time choose Days 9 and 16? Can you provide data at other times?

In conclusion, the number of experiments in this manuscript is small, and the experimental conclusions drawn with only very few experiments are unreliable. Moreover, the manuscript data is not very readable, and it is more difficult to read.

Reviewer 5 Report

Comments and Suggestions for Authors

The manuscript entitled ‘Chimeric Antigen Receptor T Cell Bearing Herpes Virus Entry Mediator Co-Stimulatory Signal Domain exhibits exhaustion resistant propertie’’ was well received.

In the manuscript, the authors have comparatively studied the impact of costimulatory domains on CAR-T cell exhaustion. There are certain things that need clarification.

The title of the manuscript has a typing error. Please rectify.

The authors conducted and analyzed the effect of different costimulatory domains on CAR-T exhaustion. The authors must delineate how their study is different from others.

Line 158-160 ‘In the absence of PD-L1, the PD-1 1+LAG3+ exhausted population of CAR-T cells was robustly elevated after first cognate antigen stimulation (Figure 3A, right side plots in the middle) but declined to almost same level of original condition after second stimulation’. Explain the reason. ? why the exhaustion increased after 1st antigenic stimulation but then decreased during second? Normally repeated antigenic stimulation leads to a more exhausted T cell phenotype.

The data in the paper needs to be more. Why the authors didn’t check CAR efficacy against a tumor model in vivo 

Comments on the Quality of English Language

fine

Round 2

Reviewer 4 Report

Comments and Suggestions for Authors

The author's answer has relieved most of my confusion, but not all of it.In Figure 6, the authors need to supplement the WB experiments to verify it, not only using flow cytometry experiments.In conclusion, the number of experiments in this manuscript is small, and the experimental conclusions drawn with only very few experiments are unreliable.

Reviewer 5 Report

Comments and Suggestions for Authors

The authors have diligently replied to all the concerns.  
